# Addressing Binary Classification over Class Imbalanced Clinical Datasets Using Computationally Intelligent Techniques

**DOI:** 10.3390/healthcare10071293

**Published:** 2022-07-13

**Authors:** Vinod Kumar, Gotam Singh Lalotra, Ponnusamy Sasikala, Dharmendra Singh Rajput, Rajesh Kaluri, Kuruva Lakshmanna, Mohammad Shorfuzzaman, Abdulmajeed Alsufyani, Mueen Uddin

**Affiliations:** 1Computer Science and Engineering, Koneru Lakshmaiah Education Foundation, Vaddeswaram 522302, India; vinodkumarfbkp@gmail.com; 2Government Degree College Basohli, University of Jammu, Basohli 184201, India; singh.gotam@gmail.com; 3New Media Technology, Makhanlal Chaturvedi National University of Journalism and Communication, Bhopal 462011, India; sasikala@mcu.ac.in; 4School of Information Technology and Engineering, Vellore Institute of Technology, Vellore 632014, India; rajesh.kaluri@vit.ac.in (R.K.); lakshman.kuruva@vit.ac.in (K.L.); 5Department of Computer Science, College of Computers and Information Technology, Taif University, P.O. Box 11099, Taif 21944, Saudi Arabia; m.shorf@tu.edu.sa (M.S.); a.s.alsufyani@tu.edu.sa (A.A.); 6College of Computing and IT University of Doha for Science and Technology, Doha P.O. Box 24449, Qatar

**Keywords:** classification, balancing techniques, clinical dataset, machine learning

## Abstract

Nowadays, healthcare is the prime need of every human being in the world, and clinical datasets play an important role in developing an intelligent healthcare system for monitoring the health of people. Mostly, the real-world datasets are inherently class imbalanced, clinical datasets also suffer from this imbalance problem, and the imbalanced class distributions pose several issues in the training of classifiers. Consequently, classifiers suffer from low accuracy, precision, recall, and a high degree of misclassification, etc. We performed a brief literature review on the class imbalanced learning scenario. This study carries the empirical performance evaluation of six classifiers, namely Decision Tree, k-Nearest Neighbor, Logistic regression, Artificial Neural Network, Support Vector Machine, and Gaussian Naïve Bayes, over five imbalanced clinical datasets, Breast Cancer Disease, Coronary Heart Disease, Indian Liver Patient, Pima Indians Diabetes Database, and Coronary Kidney Disease, with respect to seven different class balancing techniques, namely Undersampling, Random oversampling, SMOTE, ADASYN, SVM-SMOTE, SMOTEEN, and SMOTETOMEK. In addition to this, the appropriate explanations for the superiority of the classifiers as well as data-balancing techniques are also explored. Furthermore, we discuss the possible recommendations on how to tackle the class imbalanced datasets while training the different supervised machine learning methods. Result analysis demonstrates that SMOTEEN balancing method often performed better over all the other six data-balancing techniques with all six classifiers and for all five clinical datasets. Except for SMOTEEN, all other six balancing techniques almost had equal performance but moderately lesser performance than SMOTEEN.

## 1. Introduction

For the past few years, imbalanced data have attracted a significant amount of attention from learners in the machine learning area. Different challenges occur at various stages of data mining applications [1]. The development in technology and computational science has assisted the availability and growth of the data obtained from real-world problems such as medical diagnosis [2,3], credit card fault detection [4], intrusion detection, culture modeling [5], text classification, oil spill detection [6], land mine detection [7], etc., at an explosive rate [8]. A classification dataset with skewed class proportions is called imbalanced. Classifying imbalanced data is an important and frequently occurring challenge of data mining. Classes that comprise a maximum part of the dataset are known as majority classes; on the other hand, minority classes comprise a minor proportion. The major challenge that imbalanced datasets suffer is that majority of the machine learning algorithms are inclined toward the majority class. It is noteworthy that minority class has a serious concern from a learning perspective and cost significantly on misclassification [9,10,11]. Acquiring new understanding from imbalanced datasets is posing a new challenge for various data mining applications. This challenge reveals itself in two forms: minority interests and uncommon examples [12,13]. Standard learning algorithms have to compromise their performance while dealing with imbalanced learning problems [14]. It has been proved by most of the state-of-the-art classifiers that biased class distribution is the major reason for the significant loss of performance which is demonstrated by the imbalance ratio (IR) is the ratio of the number of instances in the majority class to the number of instances in the minority class. Many algorithms are employed to get rid of class imbalance problems, such as data sampling and boosting [15,16]. Data sampling has its own merits and demerits in terms of time safety and information loss. In various applications of supervised learning, a substantial difference among the prior probabilities of different classes is absorbed. The condition is known as the imbalance problem of class [17]. Most machine learning algorithms have faced challenges in countering the problem of classification of imbalanced data [18,19,20]. Data imbalance is the result of the nature of dataspace. The summarized details of various significant clinical datasets are presented in Table 1. Imbalance data classification is one of the top ten challenging issues of data mining [21]. The medical datasets often face the problem of imbalance. Herein, we used five clinical datasets for our study. In women, after skin cancer, breast cancer is the second most common cancer. In 2018, World Health Organization informed 2.09 million persons suffering from breast cancer, and 627,000 died because of this disease. It develops in breast cells, and females are the major sufferers than males. A block in the breast, discharge (bloody) from the nipple and breast shape changes are the main symptoms [22]. Coronary Heart Disease (CHD) grows in a condition where arteries are unable to supply sufficient oxygen-rich blood to the heart. Generally, it is caused due to the plague (a waxy substance) building up in the larger coronary arteries, and consequently, the flow of the blood in larger arteries is blocked. In 2017, CHD, a very common heart disease, killed 365,914 people. About 20% of deaths due to CHD are in adults below 65 years of age [23]. Liver disease causes almost 2 million deaths in a year across the globe. Some of the causes of liver disease are alcohol, obesity, viruses, or it can be inherited genetically. A deadly condition where the liver is failed by the scarring (cirrhosis) result of the damaged liver) [24]. Coronary kidney disease (CKD) means the kidneys are unable to filter the blood. Persons with high blood pressure or diabetes are at higher risk for kidney disease. High blood pressure and heart disease are the results of extra water and waste in the body caused due to the malfunctioning of the kidney. As per the study, 37 million people, which is around 15% of US adults suffering from CKD and 90% of the adults with CKD, are unaware of it, and 50% of the persons who are at low kidney function are not aware of the CKD if they are not at dialysis. According to current estimates: CKD is more common in the age group of 65 years or older (38%) than in persons of the age group of 45–64 years (13%) or 18–44 years (7%), women (15%) are badly sufferer than men who are 12% with CKD). Dialysis and kidney transplant are the treatments for kidney failure [25,26]. Diabetes is a chronic disease and is caused when insulins are not produced by the pancreas or the insulin produced is not properly used in the body. The occurrence of diabetes in 2019 is assessed to be 9.3% (463 million people) globally, amounting to 10.2% (578 million) by 2030 and 10.9% (700 million) by 2045. The effect is higher in urban areas (10.8%) than in rural (7.2%) areas and in rich (10.4%) than poor countries (4.0%), and 50.1% of persons suffering from diabetes are not aware of having the disease. The prevalence of impaired glucose tolerance is assessed to be 7.5% (374 million) in 2019 globally and is predicted to reach 8.0% (454 million) by 2030 and 8.6% (548 million) by 2045 [26]. Worldwide, lung cancer remains the major reason for the deaths of women and men suffering from cancer. Worldwide, the third most common cancer is lung cancer. The uncontrolled growth of abnormal cells in one or both lungs leads to lung cancer. The abnormal cells are unable to function normally and don’t grow into healthy lung tissue. With the growth of abnormal cells, the tumors can be formed and obstruct the normal function of the lungs, which supplies oxygen to the body via the blood. World Health Organization reported 1.76 million deaths out of 2.09 million total cases of lung cancer in 2018, and 10% of the deaths in cancer are due to lung cancer. The survival of lung cancers is decided by the stage of the diagnosis. Survival is poorer if diagnosed at a late stage [27].

In this paper, the seven algorithms are used for balancing the imbalanced data over the five clinical datasets. The six well-known classifiers are implemented to classify the data. To evaluate the performance, the four parameters—accuracy, precision, recall, and F1—score are used in this study. What is imbalanced? The response ranges from mild to extreme, as shown in Table 2. The imbalance ratio (IR) for binary class data is the ratio of number of samples of the majority class to the number of samples of the minority class.
IR = (No. of samples in Majority Class)/(No. of samples in Minority Class)(1)

Class imbalance learning approaches can be divided into three major categories: (1) data-level strategy, (2) algorithm-level strategy, and (3) hybrid strategies as shown in Figure 1. At the data-level strategy, the resampling procedure is used to handle class imbalance issues in imbalanced datasets. Further, the data-level strategy is divided further into random undersampling, oversampling, and the hybrid approach, which is a combination of undersampling and oversampling. For dealing with imbalanced data, an algorithm-level strategy may develop or update current algorithms and evaluate the consequences of minor classes. The hybrid strategy combines both data-level strategy and algorithm-level strategy to deal with the class imbalance problem.

The data level strategy for balancing the class data is more successful, and it is implemented prior to the learning process during the data preprocessing stage. Hence, the main contribution of this paper is to design a performance evaluation setup and analyze the performance effects of important data-balancing techniques with various classification methods on five imbalanced clinical datasets: Breast Cancer Disease, Coronary Heart Disease, Indian Liver Patients, Pima Indians Diabetes Database, and Coronary Kidney Disease.

The paper is organized as follows: Section 2 outlines the related work dealing with the imbalanced data. Section 3 of this paper discusses the various algorithms used for balancing the clinical data. Section 4 talks about the experimental setup and gives a description of the dataset. The results are discussed in Section 5 of this paper. The conclusion is discussed in Section 6.

## 2. Related Works

In machine learning, data is crucial for training the model. In the real world, we constantly encounter the problem of imbalanced data. This section discusses the work completed towards the efficiency of some of the machine learning techniques while dealing with the different clinical datasets, as most of the clinical datasets are inherently imbalanced in nature. Various algorithms are designed to get rid of the consequences of imbalance. The very popular algorithms are studied and analyzed for the balancing of the datasets, and afterward, the different techniques of machine learning are employed to check their performances.

Undersampling and random oversampling (ROS) for majority and minority instances can ease the change of distribution for the original dataset. To conquer the downsides of the elementary sampling techniques, such as the overfitting risk involved in oversampling and menace of information loss for undersampling method, the Synthetic Minority Oversampling Technique (SMOTE) is implemented [29].

M. Mostafizur Rahman and D. N. Davis proposed a modified cluster-based under-sampling method for balancing the data, and a training set of good quality is generated for constructing classification models [17]. SMOTE offers a new technique for oversampling. The blend of undersampling and SMOTE gives better performance than plain undersampling. SMOTE was applied on various datasets having variable imbalance degree and training datasets in different amounts, which provides a diverse test field [29].

Adaptive Synthetic (ADASYN) can produce synthetic data samples adaptively for minority classes to decrease the favoritism generated by the imbalanced data distribution. Moreover, the Learning performance is improved because of the capabilities of ADASYN to change boundaries for concentrating more on tough-to-learn examples [12].

With the help of data sampling and deep neural networks, frauds can be detected in highly imbalanced data rather than big data. Random undersampling (RUS), Random oversampling (ROS), and amalgamation of the two (ROS–RUS) are implemented to learn how different class imbalance levels influence the training and performance of the model. ROS–RUS and ROS outperform RUS and baseline models with average Area Under Curve (AUC) scores of 0.8505 and 0.8509. It is confirmed from the results that when training data are imbalanced, the default decision threshold is not optimal at 0.5, and it is recommended that the threshold be used for optimizing the performance of imbalanced classes [30].

Undersampling based on clustering (SBC), here, all samples in the datasets are divided into clusters. SBC has a very fast execution time along with a high accuracy of classification in predicting the minority class samples. Sampling methods based on SBC are used to select the majority class sample from the cluster based on the distance between minority and majority class samples [31].

Applying TOMEK links as a data cleaning technique over the oversampled training set for creating better-defined class clusters. Instances from both the classes are eliminated; consequently, not only majority class examples that form TOMEK links are removed.

In the beginning, the original dataset (a) is oversampled with SMOTES (b), and then TOMEK links are acknowledged (c) and removed, generating a balanced dataset with well-defined class clusters (d). SMOTE + ENN (Edited Nearest Neighbor), the inspiration behind this method is similar to SMOTE + TOMEK links. ENN facilitates more in-depth data cleaning as ENN removes more instances than TOMEK links. Contrarily from an under-sampling method, i.e., Neighborhood Cleaning Rule (NCL), ENN is implemented to eliminate instances of both classes. consequently, instance that is misclassified by its three nearest neighbors is eliminated from the training set [32]. SMOTE has over one hundred variants [33]. Hien M. Nguyen et al. proposed a technique where the SVM is applied to the original dataset to make a distinction between the classes B-SMOTE is implemented to find the minority sample ear the hyperplane to eliminate these samples [34]. Support vector machine (SVM) was first introduced by Vapnik in 1995, and it was a great success in widespread series of applications, but while encountering imbalanced data, the performance of SVM was significantly reduced. SVM handles and works very fine with linear as well as nonlinear datasets. The important training tuples help in forming a hyperplane for defining the data separation in a higher dimensional space known as support vectors [35]. For the classification of the datasets, prominent classification techniques are used. A. Endo et al. [8] implemented seven classifiers, namely, Artificial Neural Network (ANN), Decision Trees with naive Bayes, Naive Bayes, Bayes Net, Logistic Regression (LR), ID3, J48. They proved maximum accuracy by a logistic regression model. A decision tree (DT) constructs the structure of the flow-chart; here, every node denotes a test on an attribute value, while each branch represents a result of the test work, and leaf nodes of the tree symbolize classes. In a decision tree, the classification is done with less computation, and understandable rules can be generated easily [36]. If in a dataset most of the attributes are continuous, then Gaussian Naive Bayes (GNB) is used. It is assumed in this algorithm that predictor values are samples from Gaussian distribution [37]. k-Nearest Neighbor (k-NN) prediction model is generally acknowledged as lazy learning (no learning) approach-based estimation mechanism, and it predicts on the account of k nearest numbers provided to it [37]. An Artificial Neural Network (ANN) is formed with the combination of artificial neurons which receive input, alters the internal state (activation) as per the input, and produces output [38]. From this brief literature review, it can be inferred that no single algorithm for balancing the dataset can be considered the state-of-the-art algorithm for all the datasets in all circumstances. Moreover, there is no denying the fact about not having a single machine learning technique that can be put at the top of the hierarchy in terms of performance. They can produce the best results in domain-specific applications. Summary of significant and related works from literature for balancing techniques are given in Table 3. 

## 3. Description of Data-Balancing Algorithms

The prime focus of our study is to analyze the various balancing techniques over five clinical datasets, having varying imbalance degree. In our experiment, we used seven different balancing techniques, Undersampling, ROS, SMOTE, ADASYN, SVM SMOTE, SMOTEEN, and SMOTETOMEK, for balancing the datasets. After balancing the imbalanced datasets, six machine learning techniques, LR, DT, SVM, GNB, k-NN, and ANN, are employed over Five Clinical datasets Breast Cancer Disease, Indian Liver Patient Dataset (ILPD), Kidney Disease, Coronary heart disease (CHD), and Pima Indians Diabetes.

### 3.1. Undersampling

In undersampling, the randomly selected samples are deleted from the training datasets, but random undersampling throw-outs potentially large number of samples. It could be very challenging to define the decision boundary between minority instance and majority instance because of the discarded samples, consequent upon which the performance of classification is reduced. Algorithm 1 shows the pseudo code for the undersampling approach.

**Algorithm 1:** Pseudo code for Under Sampling**Input:** Original Training dataset
   1. Select instance(x_i_) from the dataset (i.e., x_i_ ∈D)
   2. Randomly delete member (x_i_) if it belongs to the majority class in dataset
   3. Continue the process until pre-set threshold is reached
   4. Stop
**Output:** Balanced Dataset


### 3.2. Random Oversampling

In random oversampling, the samples are chosen from minority classes randomly and, with the help of replacement, are further added to the training dataset. It can be put in other ways that, in random oversampling, the instances are duplicated from minority class in the training dataset, which may result in the overfitting of some machine learning techniques. Algorithm 2 shows the pseudo code for oversampling approach.

It has been observed in many studies that random selection of samples performs quite well if not better than many processes where samples are removed intentionally. Figure 2 portrays the semantic of undersampling and oversampling strategy for class balancing.

**Algorithm 2:** Pseudo code for oversampling**Input:** Original Training dataset
   1. Select instance(x_i_) from the dataset (i.e., x_i_ ∈D)
   2. Randomly duplicate examples in the minority class
   3. Continue the process until pre-set threshold is reached
   4. Stop
**Output:** Balanced Dataset


### 3.3. SMOTE

Considering an imbalanced dataset of a very smaller number of minority samples in comparison to the majority samples, which are large in numbers, a vector space is a collection of feature vectors that represents each sample. k nearest neighbors are selected from the minority sample for every minority sample $\vec{x_{i}}$, after that $\vec{n}$ a minority sample is selected randomly. A point is chosen randomly between $\vec{n}$ and $\vec{x_{i}}$. $\vec{syn}$ is the new synthesized sample, which is further added to the dataset. Bal is the balancing parameter for controlling the synthesized samples. Bal = 1, indicates equal number of samples from minority and majority classes. G_all is the total number of samples to be synthesized while G denotes the number of samples to be synthesized from one minority sample? The synthesis of minority samples from $\vec{x_{i}}$ is repeated G times. Algorithm 3 displays the pseudo code for SMOTE [29].

**Algorithm 3:** Pseudo code of SMOTE**Input:** X (original training data), bal (balance parameter), k (number of nearest neighbors)
$1.\;S{\_}_{min}\leftarrow$ a set of minority samples in X
2. S_maj← a set of majority samples in X
$3.\;G\_all\leftarrow |S_{\_maj}|\times \;\mathrm{Bal}-\left|S{\_}_{min}\right|$
4. G←intG_allS_min
$5.\;S_{yn}\leftarrow \emptyset \;/*\;$ a set of synthesized minority samples */
$6.\;\mathrm{for}\;\mathrm{each}\vec{x_{i}}\in S{\_}_{min}$ do 
$\;\;\;\;\;6.1\;K_{i}\leftarrow k\;nearest\;neighbor\;of\;\vec{x_{i}}\;in\;S_{\_min}\;$
     6.2Forj=1 to *G* do
$\;\;\;\;\;\;\;6.2.1\overset{\leftarrow }{n}\;\leftarrow \;a\;\mathrm{sample}\;\mathrm{randomly}\;\mathrm{chosen}\;\mathrm{from}\;K_{i}$
$\;\;\;\;\;\;\;6.2.2\;\vec{diff}\leftarrow \vec{n}-\vec{x_{i}}$
$\;\;\;\;\;\;\;6.2.3\;G_{ap}\leftarrow \;\mathrm{random}\;\mathrm{value}\;\mathrm{between}\;[0,\;1]\;$
$\;\;\;\;\;\;\;6.2.4\;\vec{syn}\leftarrow \vec{x_{i}}+G_{ap}\times \vec{diff}$
$\;\;\;\;\;\;\;6.2.5\;S_{yn}\leftarrow S_{yn}\cup \left\{\vec{syn}\right\}$
      6.3 End For
7. End For 
$8.\;\mathrm{Return}\;\;X^{\prime }=X\cup S_{yn}$
Output: 

### 3.4. ADASYN

In ADASYN, more samples are generated near borderline from minority samples. r $\left[i\right]$ is the ratio of the majority samples in the k nearest neighbor of a minority sample $\vec{x_{i}}$. It calculates the likeliness of closeness to the borderline. It further is normalized for calculating $\hat{r}\left[i\right]$ and then G[i] the number of samples to be synthesized from $\vec{x_{i}}$ [12]. Algorithm 4 displays the pseudo code for ADAYSN method [13].

**Algorithm 4:** Pseudo code for ADAYSN method**Input:** X (original training data), bal (balance parameter), k (number of nearest neighbors)
1. S_min← a set of minority samples in X 
2. S_maj← a set of majority samples in X
$3.\;G\_all\leftarrow |S_{\_maj}|\times \;\mathrm{bal}-\left|S\_min\right|$
$4.\;\mathrm{For}\;\mathrm{each}\;\vec{x_{i}}\in S\_min\;\mathrm{do}$
$4.1\;NN_{i}\leftarrow k\;nearest\;neighor\;of\;\vec{x_{i}}\;in\;X$
$4.2\;r\left[i\right]\leftarrow \frac{\left|NN_{i}\cap S{\_}_{maj}\right|}{k}$
5. End for
$6.\;\mathrm{For}\;\mathrm{each}\;\vec{x_{i}}\in S\_min\;\mathrm{do}$
$6.1\;\hat{r}\left[i\right]\leftarrow \frac{r\left[i\right]}{\sum_{i}r\left[i\right]}$
$6.2\;G\left[i\right]\leftarrow \mathrm{int}\;(\hat{r}\left[i\right]\times G_{all})$
7. End For
$8.\;S_{yn}\leftarrow \emptyset$
$9.\;\mathrm{For}\;\mathrm{each}\;\vec{x_{i}}\in S\_min\;\mathrm{do}$
$9.1\;K_{i}\leftarrow k\;nearest\;neighbor\;of\;\vec{x_{i}}\;in\;S_{\_min}\;$
$9.2\;\mathrm{For}j=1\mathrm{to}G\left[i\right]\;\mathrm{do}$
$\;\;\;9.2.1\vec{n}\leftarrow \;a\;\mathrm{sample}\;\mathrm{randomly}\;\mathrm{chosen}\;\mathrm{from}\;K_{i}$
$\;\;\;9.2.2\;\vec{diff}\leftarrow \vec{n}-\vec{x_{i}}$
$\;\;\;\;\;\;9.2.3\;G_{ap}\leftarrow \;\mathrm{random}\;\mathrm{value}\;\mathrm{between}\;[0,\;1]$
$\;\;\;\;\;\;9.2.4\;\vec{syn}\leftarrow \vec{x_{i}}+G_{ap}\times \vec{diff}$
$\;\;\;\;\;\;9.2.5\;S_{yn}\leftarrow S_{yn}\cup \left\{\vec{syn}\right\}$
   9.3 End For
11. End For
$12.\;\mathrm{Return}\;X^{\prime }=X\cup S_{yn}$
**Output: *X*′ (new training data)**


### 3.5. SVM-SMOTE

In this method, the borderline area is figured out by the support vectors after training SVMs method on the original training set. Artificial data are randomly generated along the borderline linking each minority class support vector with a number of its closest neighbors. Thus, it establishes a clear boundary between minority and majority classes [34,40]. Algorithm 5 presents the pseudo code for SVM-SMOTE

**Algorithm 5:** Pseudo code of SVM-SMOTE**Borderline Oversampling (X, N, k, m)****Input:****• X:** Training set
**• N:** Sampling level (100, 200, 300, … percent)
**• k:** Number of nearest neighbors
**• m:** Number of nearest neighbors to decide sampling type (interpolation or extrapolation)
**Variables:**
$•\;SV^{+}:\mathrm{Set}\;\mathrm{of}\;\mathrm{positive}\;\mathrm{support}\;\mathrm{vectors}\;\left(\mathrm{SVs}\right)$
• T: Total number of artificial instances to be created 
•
**amount:** Array contains the number of artificial instances corresponding to each positive SV
•
**nn** Array contains k positive nearest neighbors of each positive SV
StartT ← (N/100) × |X|$\mathrm{Compute}\;SV^{+}\;\mathrm{by}\;\mathrm{training}\;\mathrm{SVMs}\;\mathrm{on}\;X\;$$\mathrm{Compute}\;\mathrm{amount}\;\mathrm{by}\;\mathrm{evenly}\;\mathrm{distributing}\;T\;\mathrm{among}\;SV^{+}$Compute nn $\mathrm{For}\;\mathrm{each}\;sv_{i}^{+}\in SV^{+},\;\mathrm{compute}\;m\;\mathrm{nearest}\;\mathrm{neighbors}\;\mathrm{on}\;X.\;$If less than a half of the m nearest neighbors come from the negative class, along the lines joining
$sv_{i}^{+}$ with its k positive nearest neighbors (in the first to *k-th* nearest neighbor order), create *amount[i]* artificial positive instances using the following formula (extrapolate to expand positive class area): $$x_{new}^{+}=sv_{i}^{+}+\rho \left(sv_{i}^{+}-nn\left[i\right]\left[j\right]\right),$$
*where nn[i][j]is the jth positve nearest neighbor of *
$sv_{i}^{+}$ σ is a random number in the range [0, 1].Otherwise, use the following formula (interpolate like in SMOTE to consolidate the current boundary area of the positive class):$$x_{new}^{+}=sv_{i}^{+}+\rho \left(nn\left[i\right]\left[j\right]-sv_{i}^{+}\right)$$$X_{new}=X\cup \left\{x_{new}^{+}\right\}$Stop
$\mathrm{Output}:\;X_{new}:\mathrm{Over}-\mathrm{sampled}\;\mathrm{training}\;\mathrm{set}$


### 3.6. SMOTEEN

Firstly, SMOTE determines the k-Nearest Neighbors (k-NNs), which is denoted by $\psi_{x_{i}}$ for each minority sample $x_{i}\in \alpha_{min}$. To generate a synthetic data sample $x_{new}$ for $x_{i}$ SMOTE randomly selects an element $\hat{x_{i}}$ in $\psi_{x_{i}}$ and $\hat{x_{i}}$ in $\alpha_{min}$. The feature vector of $x_{new}$ is the sum of the feature vectors of $x_{i}$ and the value, which can be obtained by multiplying the vector difference between $\hat{x_{i}}$ $\mathrm{and}\;x_{i}$ and a random value δ which is between 0 and 1. By doing so, we obtain a synthetic point along the line segment joining $x_{i}$ and $\hat{x_{i}}$. Further, the Edited Nearest Neighbour (ENN) is applied to clean the overlapping of classes. Algorithm 6 contains the pseudo code for SMOTEEN [33].

**Algorithm 6:** Pseudo code of SMOTEEN**Input: Imbalance Training Data**
 1. $\mathrm{Randomly}\;\mathrm{select}\;x_{i}\;\mathrm{in}\;\mathrm{minority}\;\mathrm{classes}$
 2. $\mathrm{Identify}\;k-\;\mathrm{nearest}\;\mathrm{neighbor}\;\mathrm{of}\;x_{i}\;:\;\psi_{x_{i}}$
 3. Generate $x_{new}=x_{i}+(\hat{x_{i}}-x_{i})\times \delta$

 4. Does balancing ratio satisfy if no goto 1     else 
 5. Remove noise sample using ENN 
     ENN {
   For every observation O   Find the three nearest neighbors of O    If O gets misclassified by its three nearest neighbors
    Then delete O
    End IF
   End For
    }
 6. End
**Output:** Balanced Training Data

### 3.7. SMOTETOMEK

It is another modified version of SMOTE, where the TOMEK links are used for removing the noisy data. The TOMEK links are defined as if instance l is the nearest neighbor of instance m and m is the nearest neighbor of l, further l and m belong to different classes [32]. Algorithm 7 shows the pseudo code for SMOTE.

**Algorithm 7:** Pseudo code of SMOTETOMEK**Input: Imbalance Training Data**
 1. $\mathrm{Randomly}\;\mathrm{select}\;x_{i}\;\mathrm{in}\;\mathrm{minority}\;\mathrm{classes}$
 2. $\mathrm{Identify}\;k-\;\mathrm{nearest}\;\mathrm{neighbor}\;\mathrm{of}\;x_{i}\;:\;\psi_{x_{i}}$
 3. $\mathrm{Generate}\;x_{new}=x_{i}+(\hat{x_{i}}-x_{i})\times \delta$
 4. Does balancing ratio satisfy IF No goto 1   Else
 5. Remove noise sample using TOMEK
   TOMEK (*l, m*)
   { 
    *l* is the nearest neigbhour of *m*. 
    *m* is the nearest neigbhour of *l*. 
    *l* and m belong to different classes. 
    }
 6. End
**Output:** Balanced Training Data


## 4. Description of Classification Methods

An explanation in brief for every classification technique implemented in this study is given below so as to give the fundamental information regarding these classification methods:

### 4.1. Logistic Regression

Logistic regression yields probabilistic approximations rather than predictive analysis [41,42]. The relation between one or more variables (independent) is described and is also used for explaining the data. In more simple terms, it presents a model that gives a probability of events happening as a linear function of a set of predictor variables. The estimated regression model can be represented by Equation (1)
(2)$$\hat{p}=\frac{e^{\beta 0+\beta 1x1}}{1+e^{\beta 0+\beta 1x1}}$$

### 4.2. Decision Tree

A flow-chart-like tree structure, wherein every internal node represents a test on an attribute, every branch gives an outcome of the test, and class distribution is represented by a leaf node is classed as a decision tree. The peak node in a tree is called the root node. A decision tree can produce understandable rules easily and performs classification in lesser computation [43]. It is shown in Figure 3.

### 4.3. Support Vector Machine

A very powerful and widespread mechanism of classification was developed by V. Vapnik [44]. A division between two data levels is made with a hyperplane, and these two data levels fall on both sides of the hyperplane. The effort is always made to maximize the margin and thereby to make the sufficient probable gap amid the instances and segregating the hyperplane on either side of it.

Equation (3) is a representation of segregating hyperplane.
(W • X + b = 0)(3)

Here, W = {$\;w_{1},\;w_{2},w_{3}\ldots \ldots w_{n}$} represents the weight vector, X is n-dimensional vector, ‘n’ is number of attributes, and ‘b’ stands for a scalar (a bias) [43].

For a given dataset D = {($x_{i},y_{i}\;/x_{i}\in R_{n},\;y_{i}\in \left(-1,1\right)$),
[W • Xi + b ≥ 1](4)
for $y_{i}\;$= 1 (Label: class 1)
[W • Xi + b ≤ −1] (5)
for $y_{i}\;$= −1 (Label: class −1).

### 4.4. k-Nearest Neighbour

k-Nearest Neighbor (k-NN) prediction model is generally acknowledged as lazy learning (no learning) approach-based estimation mechanism, and it predicts on account of k nearest numbers provided to it. Generally, the neighborhood is measured using the Euclidian distance formula [37], but as per the requirement, other distance measures such as Minkowski, Hamming, and Manhattan distances are also used [43]. The distance between two points *x* and *y* is measured by the formula given by the Equation (6).
(6)$$\mathrm{dist}\left(x,\;y\right)=\sqrt{\sum_{i=0}^{n}{\left(xi-yi\right)}^{2}}$$

### 4.5. Gaussian Naïve Bayes

Gaussian Naïve Bayes is used if most of the attributes in the examples are continues. The conditional probability is given by the formula given in Equation (7):(7)$$p\left(x_{i}|y\right)=\frac{1}{\sqrt{2\prod \sigma^{2}}}\;\exp(\frac{\;{\left(x_{i}-\mu_{y}\right)}^{2}}{2\sigma_{y}^{2}})$$
where $\mu_{y}$ and $\sigma_{y}$ are mean and variance of predictor distribution.

### 4.6. Artificial Neural Network

Artificial Neural Networks simplify and imitate the brain behavior. ANN is a network of modules known as artificial neurons which receive input, vary their internal state (activation) in line with that of input, and produce output as per the input and activation [38,43].

**Weights (W):** {$\;w_{1},\;w_{2},w_{3}\ldots \ldots w_{n}$} represents the neuron strength.

**Bias (b):** It aids in the modification of the curve of the activation function.

**Input Layer:** The input layer incorporates inputs and weights.

**Activation Function:** A very important part is activation function, which gives nonlinear characteristics to the neural networks. It mainly converts any input of an artificial neuron (AN) as output. Thereafter, the obtained output is served as input to the next layer of AN [45,46]. There are many activation functions, such as the sigmoid function Equation (8).
(8)$$\mathrm{Sigmoid}\left(x\right)=\frac{e^{x}}{1+e^{x}}=\frac{1}{1+e^{-x}}$$

**Hidden Layer:** Many hidden layers may be there in ANN. Basically; hidden layer has both summation as well as activation function.

**Output Layer:** The output layer has the set of outcomes generated by the preceding layer.

## 5. Performance Metrics of Classifiers

**Confusion Matrix (CM):** The confusion matrix is a tabular representation that describes the brief assessment of the performance of a classification model [43]. The diagonal values are ones where the learning algorithm gives the correct results.

**True Positive (TP):** The training instances of which the true class is positive and which also have been positively hypothesized by us. They can be called true positives.

**False Positive (FP):** Those training instances which are negative but wrongly classified as positive by learning algorithm.

**True Negative (TN):** The training instances which are actually negative and are also hypothesized as negative.

**False Negative (FN):** The training instances are positive, but the learning algorithm is classifying these instances wrongly as negative.

**Accuracy:** -It is defined as the proportion of all true results to the total number of cases checked.
(9)$$Accuracy=\frac{\left(TP+TN\right)}{\left(TP+TN+FP+FN\right)}$$

**Precision:** Precision speaks about how trustable is the model prediction.
(10)Precision=TPTP+FP

**Recall:** Ability of the model to detect the class
(11)$$\mathrm{Recall}=\;\left(TP\right)⁄\left(TP+FN\right)$$

**F-Score/F-Measure:** It combines the precision and recall for the assessment of the classifier.
(12)$$F1-Score=2*\frac{\left(precision*recall\right)}{\left(\;precision+recall\right)}$$

It can be put in a more simplified way:Accuracy alone is not a sufficient metric to evaluate a classification model time it is misleading.High recall and high precision—This is a good model.Low recall and high precision—Model cannot detect the classes, but it is highly trustable when it does.High recall and low precision—Model can detect the classes but includes points of other classes in it.Low recall and precision—Poor model.

## 6. Experimental Setup

To accomplish the goal of comprehensive empirical performance analysis of different classifiers with several data-balancing techniques over the clinical datasets, the experiments were conducted to evaluate the efficiency and effectiveness of the algorithms in terms of classifier accuracy (CA), precision, recall, F1 score/F measure. The whole experiment was conducted using python programming language on the ‘Google Colab’ environment that runs entirely in the cloud. Figure 4 depicts the experimental workflow of the proposed work.

### Clinical Datasets

The clinical datasets are medical records collected from different patients for a specific disease. The clinical datasets are beneficial for providing cost-effective solutions for healthcare and medical diagnosis software systems. The five clinical datasets, Breast Cancer Disease, Indian Liver Patient, Coronary Kidney Disease, Coronary Heart Disease, and Pima Indians Diabetes Database, under this study have been downloaded from the UCI Machine Learning repository and detailed with their set of features, instances, imbalance ratio (IR), degree of imbalance in Table 4.

## 7. Results and Discussion

The experiments have been conducted for the review of seven balancing techniques and six classification techniques over five class imbalanced clinical datasets, as described in Table 4. Figure 5a–e demonstrates the effect of applying the various data-balancing methods. To assess the results of classification, the evaluation has been performed on the basis of well-known performance measures, namely Accuracy, Precision, Recall, and F1 score.


**Breast Cancer Disease dataset**


The breast cancer disease dataset was first preprocessed, and then each of the seven data-balancing procedures—undersampling, random oversampling, SMOTE, ADASYN, SVM-SMOTE, SMOTEEN, and SMOTETOMEK—was applied separately. As illustrated in Figure 4, the balanced dataset was then tested against six significant classifiers. The following observations were noted:The balancing technique SMOTEEN with k-NN, SVM, LR, and ANN shows the accuracy of 99.8%, 99.5%, 99.1%, and 99.1%, respectively. There is a 3% increase in the accuracy as compared to classification without data imbalance (Refer to Figure 6).Precision value for both SVM and ANN with SMOTEEN was reported as 100%. LR and k-NN also show a comparable precision value of 99.5% (Refer to Figure 7).Recall varies from 97.2 to 100% for all classifiers in general when SMOTEEN was applied. SVM reported the 100% recall for the BCD dataset (Refer to Figure 8).F1 Score for k-NN, SVM, and ANN with SMOTEEN observed 99.8, 99.5, and 99.1%, respectively (Refer to Figure 9).Thus, the balancing technique SMOTEEN for BCD provides the highest accuracy, Recall, Precision, and F1 score over all the Machine learning techniques, especially k-NN outperforms all others.


**Indian Liver Patient Dataset**


The **ILPD** dataset was also experimented with as BCD dataset. The following observations were seen-
SMOTEEN for ILPD SMOTEEN as compared to other six data-balancing techniques– Undersampling, ROS, SMOTE ADASYN, SVM-SMOTE, and SMOTETOMEK give high Accuracy 89.4%, 89.4, 86.6%, 86.1 with k-NN, DT, GNB, and LR respectively (Refer to Figure 10).Undersampling underperforms with all the classification methods due to loss in significant data while data balancing in ILPD.SMOTEEN as compared to the other six data-balancing techniques shows better precision for GNB, DT, KNN, LR, and ANN with 94.8%, 89.5%, 89.5, 89.5%, and 88.5%, respectively (Refer to Figure 11).Likewise, recall for k-NN and DT was 86.7% and for LR it is 83.7% with SMOTEEN, whereas SVM, GNB, and ANN give low values.F1 score for all machine learning techniques with SMOTEEN as a balancing technique also gives a high recall value of 88.1% for both k-NN and DT (refer to Figure 12), whereas LR, GNB, and ANN give a poor performance with low F1-score values, i.e., 84.5%, 83.4%, and 78.4%, respectively (refer to Figure 13).Thus, the experimental analysis recommends the balancing technique SMOTEEN with k-NN is the most suitable for detecting liver disease compared to the other six balancing techniques. Moreover, SMOTEEN with Decision Tree (DT) also projected considerably equal performances for ILPD Dataset.


**Coronary Kidney Disease Dataset**


When Coronary Kidney Disease dataset was experimented as BCD and ILPD dataset, the following observations were noticed:SMOTE gives the highest value of Accuracy, i.e., 99.2% on LR and 98.4% on DT, while ROS gives the highest value of 98.4% on SVM model, SMOTEEN gives the highest value 98.2% over GNB, 96.9% over LR, 95.7% over SVM, and 94.5 over k-NN, respectively (refer to Figure 14).ROS has outperformed all the balancing techniques over all the machine learning algorithms while measuring precision (refer to Figure 15).Recall for the kidney disease dataset is highest for SMOTE over LR (99.2%), DT (99.2%) and SVM (100%) (Refer to Figure 16) machine learning models, but recall value is highest for undersampling technique over GNB, Highest for SMOTEEN over k-NN and ANN gives the best result over imbalanced data without any balancing technique.ROS as compared to the other six data-balancing techniques shows better precision for GNB, DT, LR, SVM, ANN, and k-NN with 100%, 99.2%, 99.2%, 99.2%, 98%, and 90.8%, respectively.F1 score is highest for SMOTE over LR, DT, and SVM, giving the highest value of 99.2%, 98.4%, and 96.9%, respectively, whereas SMOTEEN gives the highest value of 97.9% over GNB and 96.5 for LR (refer to Figure 17).


**Coronary Heart Disease dataset**


When the **CHD** dataset was experimented, the following observations were noticed-

k-NN gives the highest value of accuracy, i.e., 92.2% for SMOTEEN, and DT gives 84% for SMOTEEN as compared to all other classifiers and balancing techniques (refer to Figure 18).SMOTEEN gives the highest value of 90% precision for k-NN, but DT, GNB, and SVM are also found to be better (refer to Figure 19).SMOTEEN gives the highest value of recall, 98.6% over k-NN but GNB and ANN underperform over CHD (refer to Figure 20).SMOTEEN reported the highest F1 Score value of 94.1%, whereas classifiers DT, SVM, and LR with SMOTEEN displayed an F1 Score of 87.6%, 82.8%, and 82.5%, respectively (refer to Figure 21).


**Pima Indians diabetes dataset**


When the diabetes dataset was experimented with the proposed experimental setup, the following observations were noticed-

SMOTEEN for k-NN, SVM, DT, LR, GNB, and ANN attains the accuracy of 96.2%, 92.5%, 91.3%, 90.6%, 87.5%, and 85.7%, respectively (Refer to Figure 22), whereas all other six data-balancing techniques underperform in terms of accuracy with all six classifiers over the Diabetes dataset.Precision values for k-NN and SVM with SMOTEEN displayed 94.8% and 93.9% (Refer to Figure 23).k-NN with SMOTEEN yields a recall of 98.6% over the diabetes dataset (Refer to Figure 24.F1 score for k-NN, SVM, DT, LR, GNB, and ANN yields 96.7%, 93.2%, 92.4%, 91.8%, 88.4%, and 87.9%, respectively (Refer to Figure 25).By and large, k-NN with SMOTEEN outperforms diabetes datasets compared to all other balancing and techniques and all other classifiers.

It is quite evident from the result analysis that the SMOTEEEN balancing method often performed better over all the other six data-balancing techniques for all five clinical datasets. This is because SMOTEEN combines oversampling and under-sampling with SMOTE and Edited Nearest Neighbors. Additionally, ENN leans towards removing a larger number of instances as compared to the Tomek links. ENN works for the elimination of cases in all classes, so any case which undergoes misclassification from all three nearest neighbors will be disposed of in the training set. In many cases, undersampling underperformed because it had discarded potentially useful instances from clinical datasets.

ROS also underperformed with different classifiers because of making exact copies of existing examples which posed overfitting to the model.

SMOTE moderately underperformed in some cases as compared to SMOTEEN because of the lack of flexibility and overgeneralization done by it. It does not just replicate the present minority cases as an alternative; SMOTE takes instances of feature space for each target class and its neighbors and then makes new instances that syndicate the attributes of the target cases with attributes of its neighbors.

ADASYN is a slight improvement over SMOTE by adding a random small value to the points to make it more genuine.

The main attention of SVM-SMOTE was on producing the new minority class samples near the dividing line with the SVM approach to support establishing the borderline between classes. Thus, wherever overfitting did not occur, the SVM-SMOTE gave a comparable result. Opposite class paired instances that are the closest neighbors to each other come under the Tomek links. Hence, the majority of the class instances from these links are eliminated as it is thought to rise the class segregation close to the decision boundaries. Therefore, in place of removing the instance solely from the majority class, in general, instances are removed from both the classes from the Tomek links. Consequently, sometimes inappropriate operation causes poor results.

## 8. Conclusions

The classification of data into specified class labels has always been a great challenge, and it is even more persistent while dealing with imbalanced data. In this study, we have implemented seven balancing techniques—Undersampling, Random oversampling, SMOTE, ADASYN, SVM-SMOTE, SMOTEEN, and SMOTETOMEK—and six different disease predication techniques—Logistic regression, Decision Tree, Support Vector Machine, k-Nearest Neighbor, and Artificial Neural Network—over five different clinical datasets, namely BCD, ILPD, CKD, CHD, and Pima Indians Diabetes Database.

SMOTEEN with k-NN provided the highest accuracy, Recall, Precision, and F1 score over all the machine learning techniques all others for the BCD dataset and bagged a 3% increase in the accuracy as compared to classification without data imbalance.SMOTEEN with k-NN was found the most suitable for detecting liver disease.Moreover, k-NN gives the highest value of accuracy of 92.2% over coronary heart disease for SMOTEEN compared to all other classifiers and balancing techniques.As for as the diabetes dataset is concerned, SMOTEEN with k-NN was found the most suitable, with accuracy of 96.2.SMOTE with Logistic regression (LR) gives the highest value of accuracy, 99.2%, over the CHD dataset.

The performance of these balancing algorithms has been observed and it is concluded that there is no single balancing technique that can generate the best results over all the datasets. If dataspace is important, then machine learning techniques cannot be ignored, and the balancing algorithms are equally important.

## Figures and Tables

**Figure 1 healthcare-10-01293-f001:**
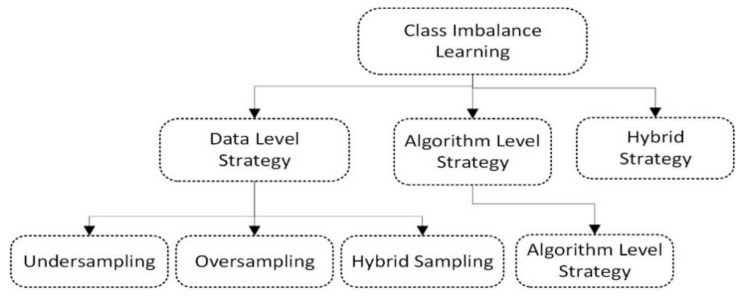
Categorization of class imbalance learning.

**Figure 2 healthcare-10-01293-f002:**
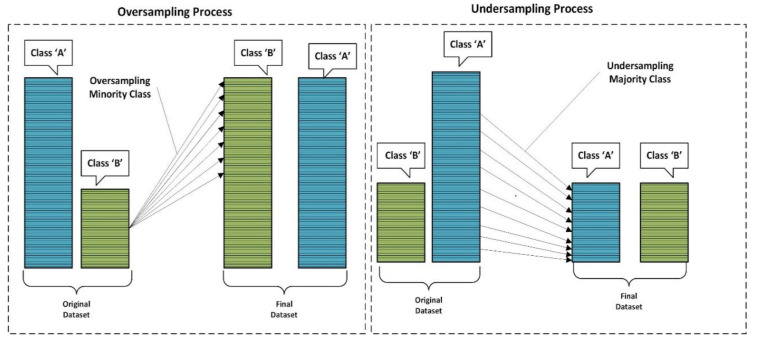
Oversampling and undersampling process.

**Figure 3 healthcare-10-01293-f003:**
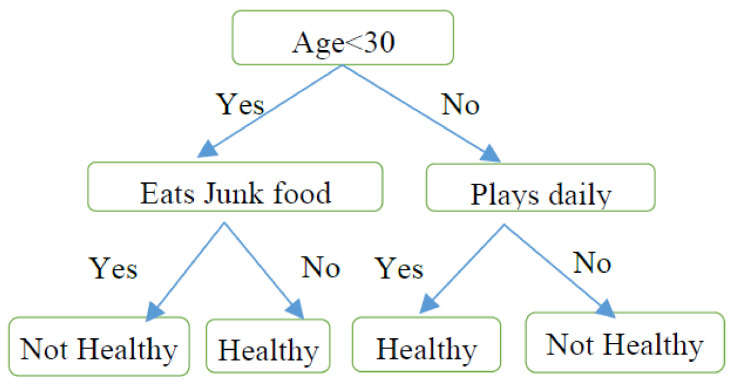
Decision making in personal health.

**Figure 4 healthcare-10-01293-f004:**
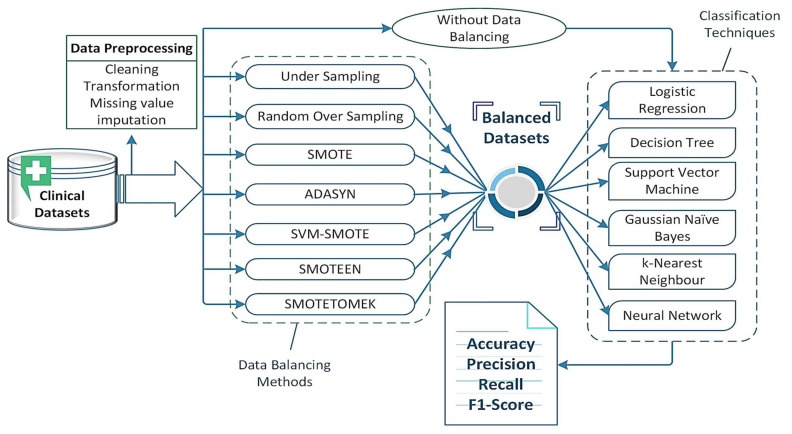
Experimental setup for evaluation of classifiers over clinical datasets.

**Figure 5 healthcare-10-01293-f005:**
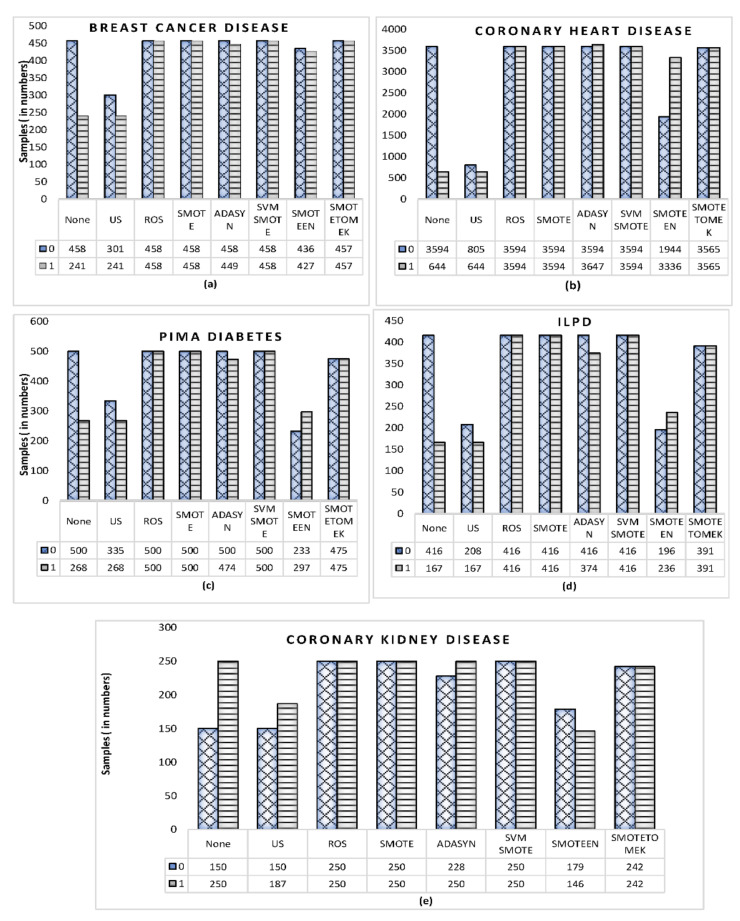
Class label counts before and after applying the various data-balancing techniques for datasets: (**a**) Breast cancer disease, (**b**) Coronary heart disease, (**c**) Pima diabetes dataset, (**d**) Indian liver patient disease, (**e**) Coronary kidney disease.

**Figure 6 healthcare-10-01293-f006:**
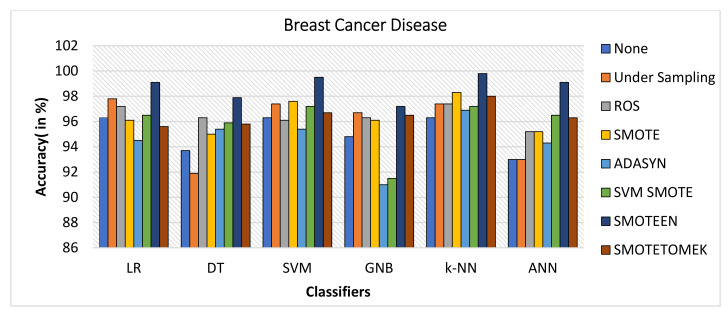
Accuracy for breast cancer disease dataset of balancing algorithms over six classifiers: Logistic regression (LR), Decision tree (DT), Support vector machine (SVM), Gaussian Naïve Bayes (GNB), k-Nearest Neighbor(k-NN), Artificial Neural Network (ANN), Random oversampling (ROS), Synthetic Minority Oversampling Technique (SMOTE),Adaptive Synthetic (ADASYN), SMOTE-Edited Nearest Neighbor (SMOTEEN).

**Figure 7 healthcare-10-01293-f007:**
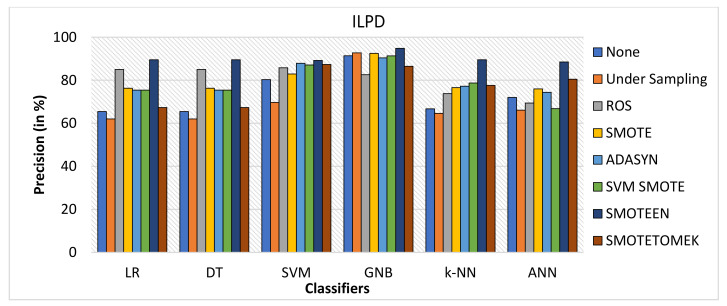
Precision for breast cancer disease dataset of balancing algorithms over six classifiers.

**Figure 8 healthcare-10-01293-f008:**
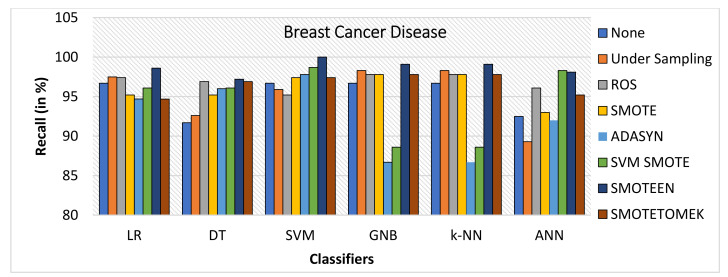
Recall for breast cancer disease dataset of balancing algorithms over six classifiers.

**Figure 9 healthcare-10-01293-f009:**
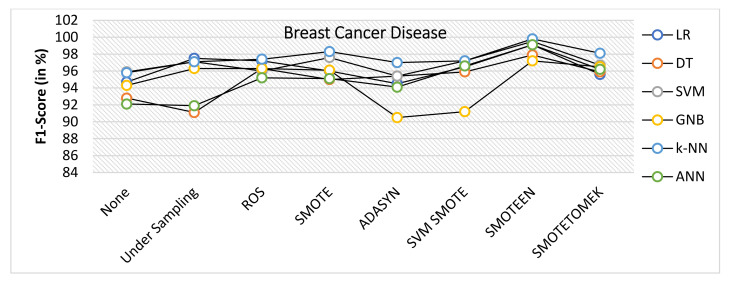
F1 Score for breast cancer disease dataset of balancing algorithms over six classifiers.

**Figure 10 healthcare-10-01293-f010:**
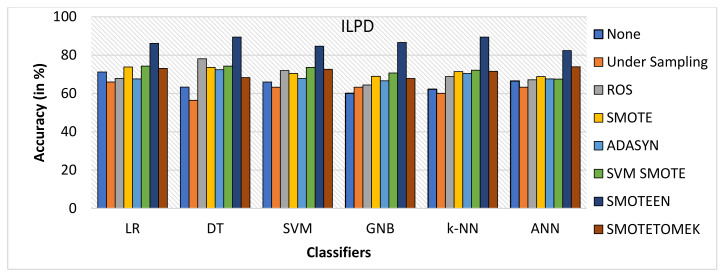
Accuracy for ILPD dataset of balancing algorithms over six classifiers.

**Figure 11 healthcare-10-01293-f011:**
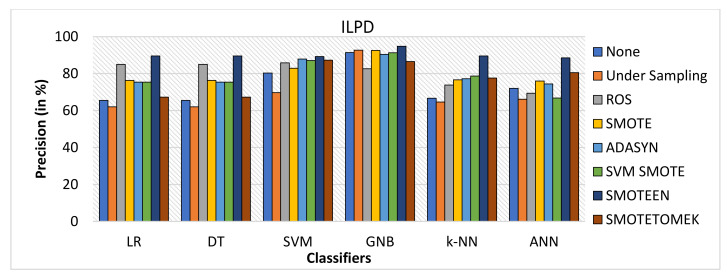
Precision for ILPD dataset of balancing algorithms over six classifiers.

**Figure 12 healthcare-10-01293-f012:**
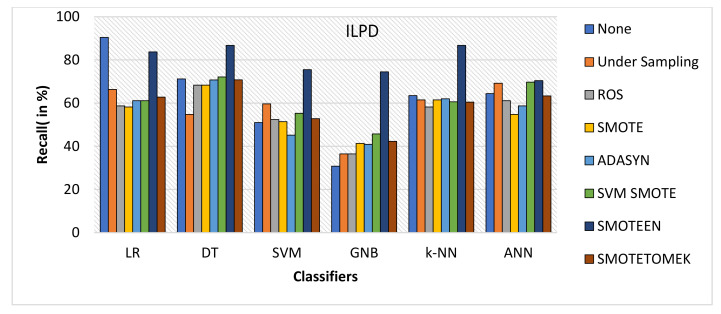
Recall for ILPD dataset of balancing algorithms over six classifiers.

**Figure 13 healthcare-10-01293-f013:**
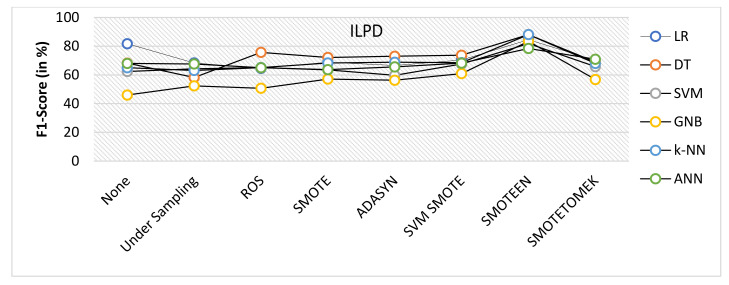
F1 score for ILPD dataset of balancing algorithms over six classifiers.

**Figure 14 healthcare-10-01293-f014:**
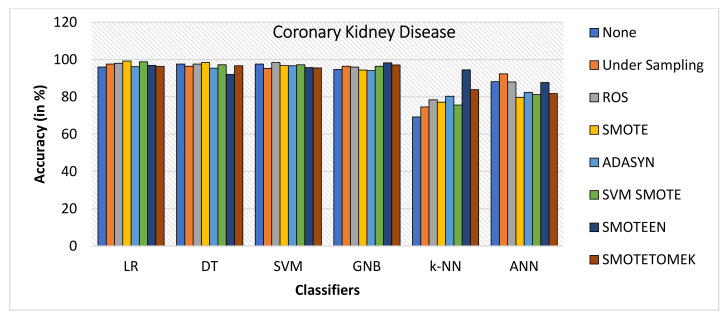
Accuracy for kidney disease dataset of balancing algorithms over six classifiers.

**Figure 15 healthcare-10-01293-f015:**
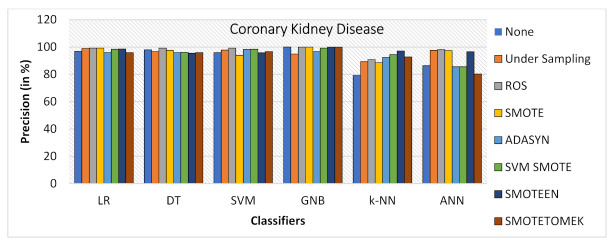
Precision for kidney disease dataset of balancing algorithms over six classifiers.

**Figure 16 healthcare-10-01293-f016:**
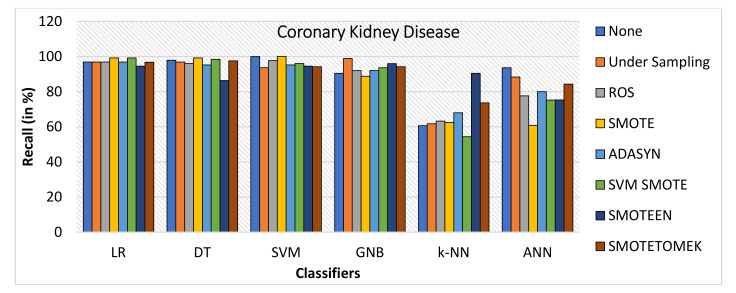
Recall for kidney disease dataset of balancing algorithms over six classifiers.

**Figure 17 healthcare-10-01293-f017:**
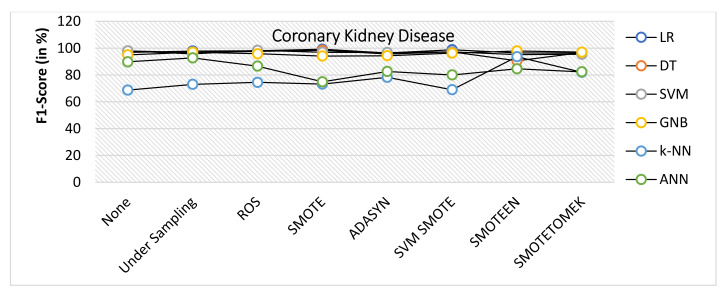
F1 score for kidney disease dataset of balancing algorithms over six classifiers.

**Figure 18 healthcare-10-01293-f018:**
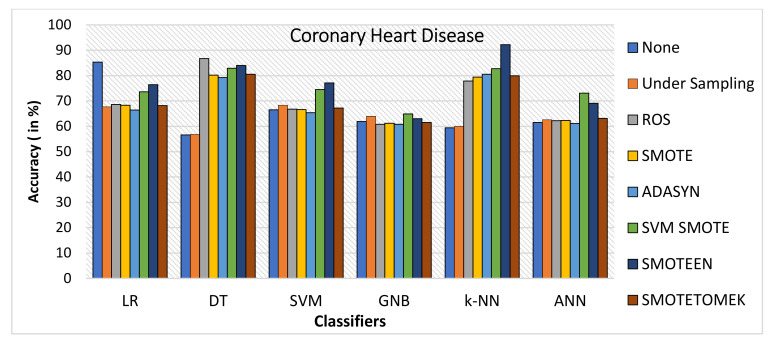
Accuracy for coronary heart disease dataset of balancing algorithms over six classifiers.

**Figure 19 healthcare-10-01293-f019:**
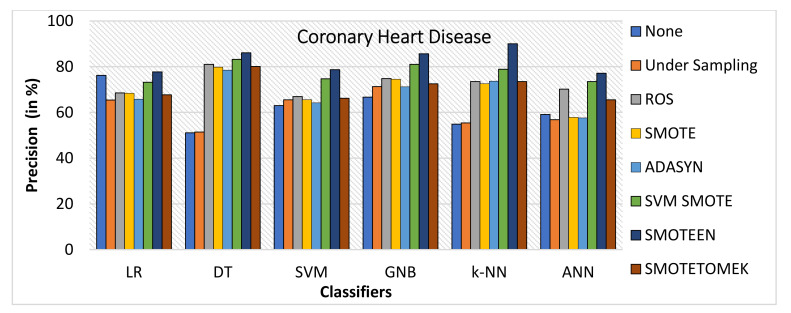
Precision for coronary heart disease dataset of balancing algorithms over six classifiers.

**Figure 20 healthcare-10-01293-f020:**
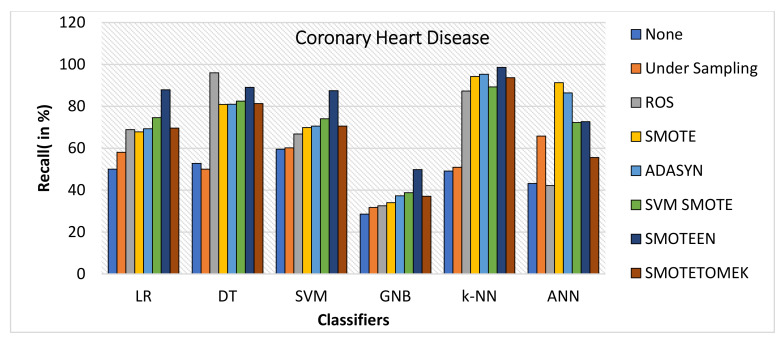
Recall for coronary heart disease dataset of balancing algorithms over six classifiers.

**Figure 21 healthcare-10-01293-f021:**
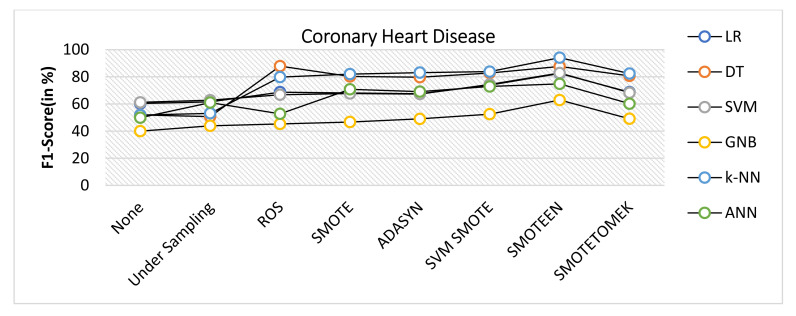
F1 score for coronary heart disease dataset of balancing algorithms over six classifiers.

**Figure 22 healthcare-10-01293-f022:**
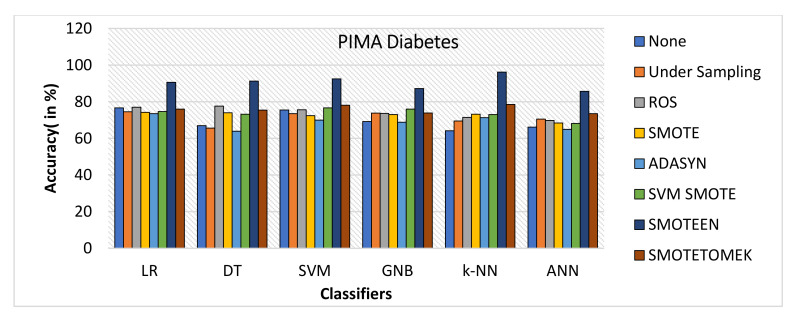
Accuracy for diabetes dataset of balancing algorithms over six classifiers.

**Figure 23 healthcare-10-01293-f023:**
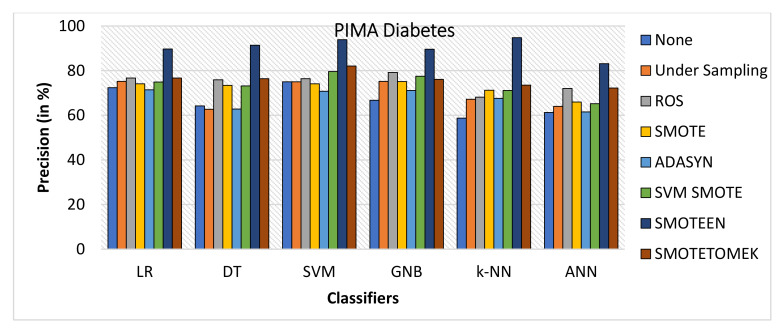
Precision for diabetes dataset of balancing algorithms over six classifiers.

**Figure 24 healthcare-10-01293-f024:**
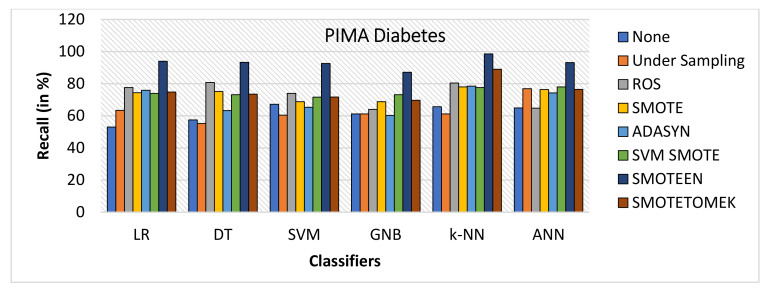
Recall for diabetes dataset of balancing algorithms over six classifiers.

**Figure 25 healthcare-10-01293-f025:**
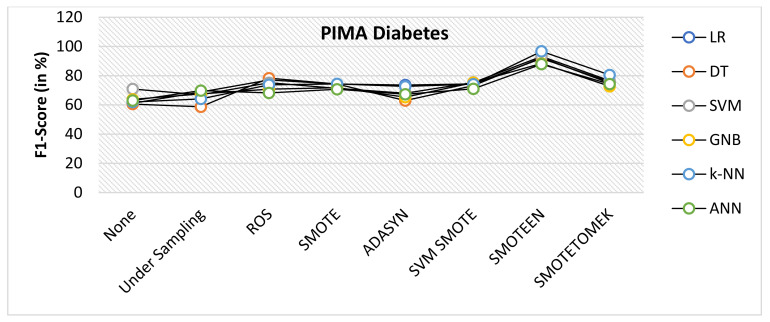
F1 score for diabetes dataset of balancing algorithms over six classifiers.

**Table 1 healthcare-10-01293-t001:** Summarized details of various clinical datasets [28].

SL	Name	Data Types	Default Task	Attribute Types	#Instances	Class Distribution	#Attributes	Imbalance Ratio
1	Breast Cancer	Multivariate	Classification	Categorical	286	**0**:201,**1**: 85	9	2.36
2	Breast Cancer Wisconsin (Original)	Multivariate	Classification	Integer	699	**0**: 458, **1**:241	10	1.9
3	Breast Cancer Wisconsin (Prognostic)	Multivariate	Classification, Regression	Real	198	**0**:151, **1**:47	34	3.21
4	Breast Cancer Wisconsin (Diagnostic)	Multivariate	Classification	Real	569	**0**:357, **1**:212	32	1.69
5	Heart Disease	Multivariate	Classification	Categorical, Integer, Real	303	**0**:164,**1**:55,2:36,**3**:35,**4**:13	75	--
6	Hepatitis	Multivariate	Classification	Categorical, Integer, Real	155	**0**:133, **1**:32	19	4.15
7	Pima Indians Diabetes Database	Multivariate	Classification	Integer	768	**0**: 500, **1**:268	8	1.9
8	Liver Disorders	Multivariate	Classification	Categorical, Integer, Real	345	**0**:145,**1**:200	7	1.37
9	Lung Cancer	Multivariate	Classification	Integer	32	**0**:23, **1**:9	56	2.55
10	SPECT Heart	Multivariate	Classification	Categorical	267	**0**:55,**1**: 212	22	3.85
11	SPECTF Heart	Multivariate	Classification	Integer	267	**0**:55,**1**:212	44	3.85
12	Thyroid Disease	Multivariate, Domain-Theory	Classification	Categorical, Real	7200	**1**:166, **2**:368, 3:6666	21	--
13	Breast Tissue	Multivariate	Classification	Real	106	**Car**:21 **Fad**:15 **Mas**:8, **Gla**:16, **Con**:14, **Adi**:22	10	--
14	Fertility	Multivariate	Classification, Regression	Real	100	**N**:88, **O**:12	10	7.33
15	Diabetic Retinopathy Debrecen Dataset	Multivariate	Classification	Integer, Real	1151	**0**:540, **1**:611	20	1.131
16	HIV-1 protease cleavage	Multivariate	Classification	Categorical	6590	**0**:5232 1:1358	1	3.85
17	Breast Cancer Coimbra	Multivariate	Classification	Integer	116	**0**:52,1:65	10	1.25
18	Parkinson’s Disease Classification	Multivariate	Classification	Integer, Real	756	**0**:192, **1**:564	754	2.94 s
19	Hepatitis C Virus (HCV) for Egyptian patients	Multivariate	Classification	Integer, Real	1385	**1**:336, **2**:332, **3**:355, **4**:362	29	--
20	Heart failure clinical records	Multivariate	Classification, Regression, Clustering	Integer, Real	299	**0**:203,**1**:96	13	2.11

The bold represents to class labels.

**Table 2 healthcare-10-01293-t002:** Classification of degree of imbalance in data.

Class Imbalance Degree	Proportion of Minority Class
Extreme	<1% of the dataset
Moderate	1–20% of the dataset
Mild	20–40% of the dataset

**Table 3 healthcare-10-01293-t003:** Summarized related works from literature for balancing techniques.

S. No	Author and Year	Techniques Applied	Claims in the Study	Cons
1	N. Chawla, K. Bowyer, L. Hall, and W. Kegelmeyer (2002) [29]	SMOTE: Synthetic Minority Oversampling Technique	An amalgamation of technique of oversampling the minority class and undersampling the majority class can attain better performance in classification. Creating synthetic minority class instances implicates the oversampling of the minority class.	Suffers from over fitting problem
2	M. Mostafizur Rahman and D. N. Davis (2013) [17]	Smote oversampling Cluster-based undersampling Modified cluster-based undersampling method	The traditional methods of balancing, such as undersampling and oversampling, may not prove to be effective and suitable over these Imbalanced datasets. The technique discussed in this paper shows better results for datasets where class level is not certain. A modified cluster-based undersampling technique produces good quality training sets in addition to balancing the datasets.	Computational costs increase
3	Haibo He, Yang Bai, Edwardo A. Garcia, and Shutao Li (2008) [12]	Adaptive Synthetic (ADASYN)	ADASYN can reduce the bias made by the imbalanced data distribution. ADASYN moves the classification decision boundary nearer to the difficult examples.	Because of its adaptability, ADASYN’s precision may degrade.
4	Justin M. Johnson, Taghi M. Khoshgoftaar (2019) [39]	ROS RUS ROS-RUS	Data sampling and deep neural networks are implemented for detecting fraud in highly imbalanced datasets.	ROS may increase the likelihood of overfitting and computational costs In RUS, sample of the majority class chosen could be biased
5	Show-Jane Yen, Yue-Shi Lee (2009) [31]	Clustering-based undersampling Clustering and distances between samples based undersampling	Back propagation neural network for imbalanced class distribution by Cluster based undersampling approaches. SBC executes fast and provides high accuracy of classification for minority classes. BCMD is stable and generates better accuracy while handling disordered and exceptional data samples.	Computational costs increase
6	G. Batista, R. C. Prati, M. C. Monard (2004) [32]	SMOTEEN, SMOTETOMEK	Random over sampling techniques gives meaningful results over other techniques at less computational rate.	SMOTE is not suitable for high-dimensional data
7	Hien M. Nguyen, Eric W. Cooper, Katsuari Kamei (2009) [34]	SVMs and B-SMOTE	This technique targets the borderline area where establishing the decision boundary is critical rather than sampling the whole of the minority class.	It could not provide big savings regarding the number of synthetically generated examples, trading to the classification accuracy

**Table 4 healthcare-10-01293-t004:** Dataset description before applying data-balancing technique.

S. No	Dataset	#Instances	#Attributes	Class	IR	Minority Class (%)	Degree of Imbalanced
1	BCD [47]	699	9	**0**:458, **1**:241	1.9	34.5	Mild
2	Chronic Heart Disease [48]	4238	14	**0**:3594, **1**:644	5.58	15.2	Moderate
3	ILPD [49]	583	9	**0**:416, **1**:167	2.49	28.6	Moderate
4	PIMA Diabetes [50]	768	8	**0**:500, **1**:268	1.86	34.9	Mild
5	Chronic Kidney Disease [51]	400	24	**0**:150, **1**:250	1.66	37.5	Mild

## Data Availability

UCI Machine Learning Repository, 2010. http://archive.ics.uci.edu/ml (accessed on 20 October 2020).

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
