# Peer review of "Addressing Binary Classification over Class Imbalanced Clinical Datasets Using Computationally Intelligent Techniques"

_healthcare, 2022, doi:10.3390/healthcare10071293_

Round 1

Reviewer 1 Report

1. May include significant results in the abstract. Balancing techniques are almost have equal performance.

2. The literature review and methodology are well explained.

3. The discussions on the results can be elaborated further.

Reviewer 2 Report

1. Author did not address problem clearly, contribution did not clearly mention.

2. lack of literature review , consider the recently work. Existing methods limitations did not well analyzed.

3. related work did not classify .figure 1 and 2 need more detail explanation.

4.Algorithm need more detail explanation each step.

5. result need more detail explanation.

Reviewer 3 Report

This paper should be extensively revised to describe each method and provide the practical use case for each method.  AS written the manuscript is not clear.  It appears that the authors have all the information that can be edited but it may take some time to complete 

Round 2

Reviewer 2 Report

All comments are well addresses.

Author Response

File Attached
